# INTERPRETING THE SECOND-ORDER EFFECTS OF NEURONS IN CLIP

**Yossi Gandelsman, Alexei A. Efros, Jacob Steinhardt**
UC Berkeley
{yossi_gandelsman,aaefros,jsteinhardt}@berkeley.edu

## ABSTRACT

We interpret the function of individual neurons in CLIP by automatically describing them using text. Analyzing the direct effects (i.e. the flow from a neuron through the residual stream to the output) or the indirect effects (overall contribution) fails to capture the neurons' function in CLIP. Therefore, we present the "second-order lens", analyzing the effect flowing from a neuron through the later attention heads, directly to the output. We find that these effects are highly selective: for each neuron, the effect is significant for $< 2\%$ of the images. Moreover, each effect can be approximated by a single direction in the text-image space of CLIP. We describe neurons by decomposing these directions into sparse sets of text representations. The sets reveal polysemantic behavior—each neuron corresponds to multiple, often unrelated, concepts (e.g. ships and cars). Exploiting this neuron polysemy, we mass-produce "semantic" adversarial examples by generating images with concepts spuriously correlated to the incorrect class. Additionally, we use the second-order effects for zero-shot segmentation, outperforming previous methods. Our results indicate that an automated interpretation of neurons can be used for model deception and for introducing new model capabilities[1].

## 1 INTRODUCTION

Automated interpretability of the roles of components in neural networks enables the discovery of model limitations and interventions to overcome them. Recently, such a technique was applied for interpreting the attention heads in CLIP (Gandelsman et al., 2024), a widely used class of image representation models (Radford et al., 2021). However, this approach has only scratched the surface, failing to explain a major set of CLIP's components—neurons. Here we will introduce a new interpretability lens for studying the neurons and use the gained understanding for zero-shot segmentation and mass-production of semantic adversarial examples.

Interpreting the neurons in CLIP is a harder task than interpreting the attention heads. First, there are more neurons than individual heads, which requires a more automated approach. Second, their direct effect on the output—the flow from the neuron, through the residual stream *directly* to the output—is negligible (Gandelsman et al., 2024). Third, most information is stored redundantly—many neurons encode the same concept, so just ablating a neuron (i.e. examining indirect effects) does not reveal much since other neurons make up for it.

The limitations presented above mean that we can neither look at the direct effect nor the indirect effect to analyze a single neuron. To address this, we introduce a "second-order lens" for investigating the *second-order effect* of a neuron—its total contribution to the output, flowing via all the consecutive attention heads (see Figure 1).

We start by analyzing the empirical behavior of second-order effects of neurons. We find that these effects have high significance in the late layers. Additionally, each neuron is highly selective: its second-order effect is significant for only a small set (about 2%) of the images. Finally, this effect can be approximated by a single direction in the joint text-image representation space of CLIP (Section 3.3).

---

[1]Project page: https://yossigandelsman.github.io/clip_neurons/.

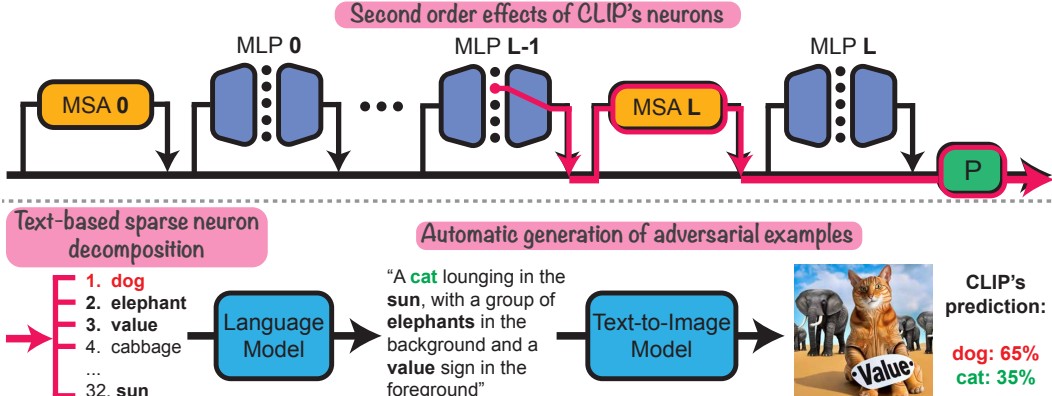

Figure 1: **Second order effects of CLIP's neurons.** Top: We analyze the second-order effects of neurons in CLIP-ViT (flow in pink). Bottom-left: Each second-order effect of a neuron can be decomposed to a sparse set of word directions in the joint text-image space. Bottom-right: co-appearing words in these sets can be used for mass-generation of semantic adversarial images.

As each direction that corresponds to a neuron lives in a joint representation space, it can be decomposed as a sparse sum of text representations that describes the neurons' functionality (see Figure 1). These text representations show that neurons are polysemantic (Elhage et al., 2022)— each neuron corresponds to *multiple* semantic concepts. To verify that the neuron decompositions are meaningful, we show that these concepts correctly track which inputs activate a given neuron (Section 4).

The polysemantic behavior of neurons allows us to find concepts that inadvertently overlap in the network, due to being represented by the same neuron. We use these spurious cues for mass production of "semantic" adversarial examples that will fool CLIP (see bottom of Figure 1). We apply this technique to automatically produce adversarial images for a variety of classification tasks. Our qualitative and quantitative analysis shows that incorporating spuriously overlapping concepts in an image deceives CLIP with a significant success rate (Section 5.1).

The text representations that describes the neurons' functionality enable an additional application— zero-shot segmentation. Mining for text representations of class names, we can identify class-relevant neurons with the second-order lens. Averaging the activation patterns of such neurons, we generate attribution heatmaps. Binarizing them yields a strong zero-shot image segmenter that outperforms recent work (Chefer et al., 2021; Gandelsman et al., 2024).

In summary, we present an automated interpretability approach for CLIP's neurons by modeling their second-order effects and spanning them with text descriptions. We use these descriptions to automatically understand neuron roles and apply this to two applications. This shows that a scalable understanding of internal mechanisms both uncovers errors and elicits new capabilities from models.

## 2    RELATED WORK

**Contrastive vision-language models.** Models like ALIGN (Jia et al., 2021), CLIP (Radford et al., 2021), and its variants (Zhai et al., 2023; Li et al., 2023) produce image representations from pre-training on images and their captions. They demonstrated impressive zero-shot capabilities for various downstream tasks, including OCR, geo-localization, and classification (Wortsman, 2023). These models' representations are also used for segmentation (Lüddecke & Ecker, 2022), image generation (Ramesh et al., 2021; Rombach et al., 2022) and 3D understanding (Kerr et al., 2023). We aim to reveal the roles of neurons in such models.

**Mechanistic interpretability of vision models.** Mechanistic interpretability aims to reverse engineer the computation process in neural networks. In computer vision, this approach was applied to model individual network components (Shah et al., 2024) and to extract intermediate mechanisms like curve detectors (Olah et al., 2020), object segmenters (Bau et al., 2019; 2020), high-frequency boundary detectors (Schubert et al., 2021), and multimodal concepts detectors (Goh et al., 2021). More closely to us, a few works made use of the intrinsic language-image space of CLIP to interpret the direct effect of attention heads and the output representation in CLIP with automatic text descriptions (Gandelsman

et al., 2024; Bhalla et al., 2024). We go beyond the output and direct effects of individual layers to interpret intermediate neurons in CLIP.

**Neurons interpretability.** The role of individual neurons (post-non-linearity single channel activations) is broadly studied in computer vision models (Bau et al., 2019; 2020; Goh et al., 2021) and language models (Radford et al., 2017; Geva et al., 2021; Meng et al., 2022). (Dravid et al., 2023; Gurnee et al., 2024) demonstrate that neurons can learn universal mechanisms across different models in both domains. Elhage et al. (2022) show that neurons can be polysemantic (i.e. activated on *multiple* concepts) and exploit this property for generation of L2 adversarial examples. Some work seeks to extract neurons' concepts by learning sparse dictionaries (Bricken et al., 2023; Rajamanoharan et al., 2024). Other methods use large language models to automatically describe neurons based on which examples they activate on (Bills et al., 2023; Oikarinen & Weng, 2023; Hernandez et al., 2022; Shaham et al., 2024). In contrast, we focus on the contribution of neurons *to the output representation*.

## 3 SECOND-ORDER EFFECTS OF NEURONS

We start by presenting the CLIP-ViT architecture. Then, we derive the second-order effect of neurons and present their benefits over first-order and the indirect effects. Finally, we empirically characterize the second-order effects, setting the stage for automatically interpreting them via text in Section 4.

### 3.1 CLIP-VIT PRELIMINARIES

**Contrastive pre-training.** CLIP is trained via a contrastive loss to produce image representations from weak text supervision. The model includes an image encoder $M_{\text{image}}$ and a text encoder $M_{\text{text}}$ that map images and text descriptions to a shared latent space $\mathbb{R}^d$. The two encoders are trained together to maximize the cosine similarity between the output representations $M_{\text{image}}(I)$ and $M_{\text{text}}(t)$ for matching input text-image pairs $(t, I)$:

$$\text{sim}(I, t) = \langle M_{\text{image}}(I), M_{\text{text}}(t) \rangle / (||M_{\text{image}}(I)||_2 ||M_{\text{text}}(t)||_2). \tag{1}$$

**Using CLIP for zero-shot classification.** Given a set of classes, each name of a class $c_i$ (e.g. the class "dog") is mapped to a fixed template $template(c_i)$ (e.g. "A photo of a {class}"), and encoded via the text encoder $M_{\text{text}}(template(c_i))$. The classification prediction for a given image $I$ is the class $c_i$ whose text representation is most similar to the image representation: $\arg\max_{c_i} \text{sim}(I, template(c_i))$.

**CLIP-ViT architecture.** The CLIP-ViT image encoder consists of a Vision Transformer followed by a linear projection[2]. The vision transformer (ViT) is applied to the input image $I \in \mathbb{R}^{H \times W \times 3}$ to obtain a $d'$-dimensional representation $\text{ViT}(I)$. Denoting the projection matrix by $P \in \mathbb{R}^{d \times d'}$:

$$M_{\text{image}}(I) = P(\text{ViT}(I)). \tag{2}$$

The input $I$ to ViT is first split into $K$ non-overlapping image patches that are encoded into $K$ $d'$-dimensional *image tokens*. An additional learned token, named the *class token*, is included and used later as the output token. As shown in Figure 1, the tokens are processed simultaneously by applying $L$ alternating residual layers of multi-head self-attention (MSA) and MLP blocks.

**MLP neurons in CLIP.** The MLP layers are applied separately on each image token and the class token. They consist of an input linear layer, parametrized by $W_{\text{in}}^l \in \mathbb{R}^{N \times d'}$, followed by a GELU non-linearity $\sigma$ and an output linear layer, parametrized by $W_{\text{out}}^l \in \mathbb{R}^{d' \times N}$. Here $l$ is the layer number and $N$ is the width (number of neurons) of the MLP. We next analyze the contributions of each individual neuron $n \in \{1, ..., N\}$ for each layer.

### 3.2 ANALYZING THE NEURON EFFECTS ON THE OUTPUT

Individual neurons have different types of contributions to the output—the first-order (direct) effects, second-order effects, and (higher-order) indirect effects. We introduce them and explain the limita-

---

[2]Throughout the paper, we ignore layer-normalization terms to simplify derivations. We address layers-normalization in detail in Appendix A.6.

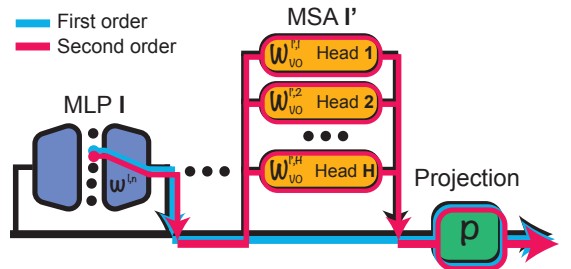
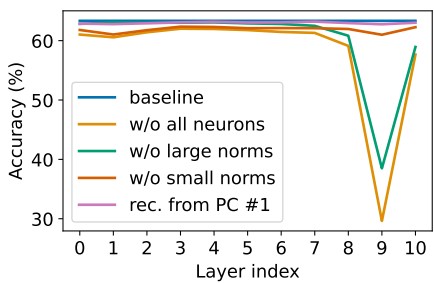

Figure 2: **First/Second-order effects.** The first order is the flow coming from a neuron to the projection layer and the output (blue). The second order goes from a single neuron through all the consecutive attention heads, to the projection layer, and to the output (pink).

Figure 3: **Mean-ablation of second order effects (ViT-B-32).** We evaluate the performance on ImageNet validation set. Second-order effects concentrate in late layers, significant for only a part of the images, and can be approximated by one direction in the output space.

tions of the direct and indirect effects before continuing to characterize the second-order effects in Section 3.3.

**First-order effects (logit lens (nostalgebraist, 2020)).** The first-order effect is the direct contribution of a component to the residual stream, multiplied by the projection layer (see blue flow in Figure 2). For an individual neuron $n$ in layer $l$, let $p_i^{l,n}(I) \in \mathbb{R}$ denote its post-GELU activation on the $i$-th token of the input image $I$. Then the contribution $e_i^{l,n}$ of the $n$-th neuron to the $i$-th token in the residual stream is:

$$e_i^{l,n} = p_i^{l,n}(I)w^{l,n} \tag{3}$$

where $w^{l,n} \in \mathbb{R}^{d'}$ is the the $n$-th column of $W_{out}^l$. As the output representation is the class token (indexed 0) multiplied by $P$, the first-order effect for neuron $n$ on the output is $Pe_0^{l,n}$.

As observed by Gandelsman et al. (2024), the first-order effects of MLP layers are close to constants in CLIP and most of the first-order contributions are from the late attention layers. We therefore focus on the second-order effects: the flow of information from the neurons through the attention layers.

**Second-order effects.** The contribution $e_i^{l,n}$ to the residual stream directly affects the input to later layers. We focus on the flow of $e_i^{l,n}$ through subsequent MSAs and then to the output (pink flow in Figure 2). We call this interpretability lens the "second-order lens", in analogy to the "logit lens".

Following Elhage et al. (2021), the output of an MSA layer $\mathsf{MSA}^l$ that corresponds to the class token is a weighted sum of its $K + 1$ input tokens $[z_0, ..., z_K]$:

$$\left[ \mathsf{MSA}^l([z_0, ..., z_K]) \right]_0 = \sum_{h=1}^{H} \sum_{i=0}^{K} a_i^{l,h}(I) W_{VO}^{l,h} z_i \tag{4}$$

where $W_{VO}^{l,h} \in \mathbb{R}^{d' \times d'}$ are transition matrices (the OV matrices) and $a_i^{l,h}(I) \in \mathbb{R}$ are the attention weights from the class token to the $i$-th token ($\sum_{i=0}^{K} a_i^{l,h} = 1$).

To obtain the second-order effect of a neuron $n$ at layer $l$, $\phi_n^l(I)$, we compute the additive contribution of the neuron through all the later MSAs and project it to the output space via $P$. Plugging in Equation (3) as the contribution to $z_i$ in Equation (4) and summing over layers, the second order effect of neuron $n$ is then:

$$\phi_n^l(I) = \sum_{l'=l+1}^{L} \sum_{h=1}^{H} \sum_{i=0}^{K} \underbrace{\left( p_i^{l,n}(I) a_i^{l',h}(I) \right)}_{\text{attention-weighted activations}} \underbrace{\left( P W_{VO}^{l',h} w^{l,n} \right)}_{\text{input-independent}} \tag{5}$$

**Indirect effects.** An alternative approach is to analyze the indirect effect of a neuron by measuring the change in output representation when intervening on a neuron's output. Specifically, the intervention is done by replacing the activation $p_i^{l,n}$ of the neuron for each token with a pre-computed per-token mean. However, as was shown by McGrath et al. (2023), models often learn "self-repair" mechanisms that can obscure the individual roles of neurons. We illustrate these issues in the next section.

| effect type | accuracy after mean-ablation | variance explained by first PC |
|---|---|---|
| indirect | 52.3 | 11.0 |
| second-order | 29.6 | 48.2 |

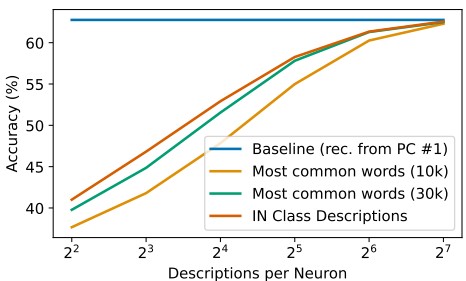

Table 1: **Comparison to indirect effect.** We compare the second-order effects and the indirect effects by mean-ablating layer 9 in ViT-B-32 on ImageNet validation set.

Figure 4: **Accuracy for neuron reconstructed from sparse text representations (ViT-B-32, layer 9).** We evaluate the sparse text decompositions for different initial description pools and description set sizes.

## 3.3 CHARACTERIZING THE SECOND-ORDER EFFECTS

We analyze the empirical behavior of the second-order effects of neurons $\phi_n^l$ derived in the previous section. We find that only neurons from the late MLP layers have a significant second-order effect and that each individual neuron has a significant effect for less than 2% of the images. Finally, we show that $\phi_n^l$ can be approximated by one linear direction in the output space. These findings will help motivate our algorithm for describing output spaces of neurons with text in Section 4.

**Experimental setting.** To evaluate the second-order effects and their contributions to the output representation, we measure the downstream performance on the ImageNet classification task (Deng et al., 2009) after ablating these effects for each neuron. Specifically, we apply mean-ablation (Nanda et al., 2023), replacing the additive contributions of individual $\phi_n^l(I)$'s to the representation with the mean computed across a dataset $D$. In our experiments, we mean-ablate all the neurons in a layer simultaneously and evaluate the downstream classification performance before and after ablation. Components with larger effects should result in larger accuracy drops.

We take $D$ to be $\sim 5000$ images from the ImageNet training set. We report zero-shot classification accuracy on the ImageNet validation set. Our model is OpenAI's ViT-B-32 CLIP, which has 12 layers. We present additional results for ViT-L-14 and for ImageNet-R (Hendrycks et al., 2021) in Appendix A.1 and Figure 10.

**Second-order effects concentrate in moderately late layers.** We evaluate the contributions of all the $\phi_n^l$ across different layers and observe that the neurons with the most significant second-order effects appear relatively late in the model. The results for different layers in ViT-B-32 CLIP model are presented in Figure 3 ("w/o all neurons"). As shown, mean-ablating layers 8-10 leads to the largest drop in performance. These layers appear right before the MSA layers with the most significant direct effects, as shown in Gandelsman et al. (2024) (layers 9-11; see Appendix A.2). The same trend is preserved for a larger model size as well (see Appendix A.1).

**The second-order effect is sparse.** We find that the second-order effect of each individual neuron is significant only for less than 2% of the images across the validation set. We repeat the same experiment as before, but this time we only mean-ablate $\phi_n^l(I)$ for a subset of images, while keeping the original effects for other images. For most of the images, except the subset of images in which $\phi_n^l(I)$ has a large norm, we can mean-ablate $\phi_n^l(I)$ without changing the accuracy significantly, as shown in Figure 3 ("w/o small norm"). Differently, mean-ablating the contributions for the 100 images with the largest $\phi_n^l(I)$ norms results in a significant drop in performance ("w/o large norm"). The same trend is shown for images from ImageNet-R in Figure 10.

**The second-order effect is approximately rank 1.** While the second-order effect for a given neuron can write to different directions in the joint representation space for each image, we find that $\phi_n^l(I)$ can be approximated by one direction $r_n^l \in \mathbb{R}^d$ in this space, multiplied by a coefficient $x_n^l(I)$ that depends on the image. We use the set $S_n^l$, which contains the largest second-order effects in norm from $D$, and set $r_n^l$ to be the first principle component computed from $S_n^l$. We approximate

| Neuron | ImageNet class descriptions | Common words (30k) |
|---|---|---|
| #4 | +"Picture with falling snowflakes" 
 +"Picture portraying a person [...] in extreme weather conditions" 
 -"Picture with a bucket in a construction site" 
 +"Photograph taken during a holiday service" | +"snowy" 
 +"frost" 
 +"closings" 
 +"advent" |
| #391 | +"Image with a traditional wooden sled" 
 +"Image with a wooden cutting board" 
 +"Picture showcasing beach accessories" 
 -"Photograph with a syringe and a surgical mask" | +"woodworking" 
 -"swelling" 
 +"cedar" 
 +"heirloom" |
| #2137 | +"Photo with a lime garnish" 
 +"Image with candies in glass containers" 
 -"Picture featuring lifeboat equipment" 
 +"Close-up photo of a melting popsicle" | +"refreshments" 
 +"gelatin" 
 +"sour" 
 +"cosmopolitan" |
| #2914 | +"Photo that features a stretch limousine" 
 +"Image capturing a suit with pinstripes" 
 +"Caricature with a celebrity endorsing the brand" 
 +"Image showcasing a Bullmastiff's prominent neck folds" | +"motorhome" 
 +"yacht" 
 +"cirrus" 
 +"cabriolet" |

Table 2: **Examples of sparse decompositions (ViT-B-32, layer 9).** We present the top-4 texts corresponding to the sparse decomposition of each neuron and the signs of the decomposition coefficients, for two initial pools ($m = 128$). See Table 5 for more neurons.

$\phi_n^l(I)$ with $x_n^l(I)r_n^l + b_n^l$, where $b_n^l \in \mathbb{R}^d$ is the bias computed by averaging $\phi_n^l(I)$ across $D$, and $x_n^l(I) \in \mathbb{R}$ is the norm of the projection of $\phi_n^l(I)$ onto $r_n^l$.

To verify that this approximation recovers $\phi_n^l(I)$ we replace each $\phi_n^l(I)$ for each neuron and image in the validation set with the approximation. We then evaluate the downstream classification performance. As shown in Figure 3 ("reconstruction from PC #1"), for each layer $l$, this replacement results in a negligible drop in performance from the baseline, that uses the full representation. The same behavior is observed for ViT-L model and for a different initial set of images in the Appendix.

**Comparison to indirect effect.** We compare the second-order effect to the indirect effect and present the variance explained by the first principle component for each of them and the drop in performance when simultaneously mean-ablating all the effects from one layer. As shown in Table 1, Mean-ablating the indirect effects results in a smaller drop in performance due to self-repair behavior. Moreover, the first principle component explains significantly less of the variance in the indirect effect, than in the second-order effect. This demonstrates two advantages of the second-order effects— uncovering neuron functionality that is obfuscated by self-repair, and one-dimensional behavior that can be easily modeled and decomposed, as we will show in the next section.

## 4 Sparse decomposition of neurons

We aim to interpret each neuron by associating its second-order effect with text. We build on the previous observation that each second-order effect of a neuron $\phi_n^l$ is associated with a vector direction $r_n^l$. Since $r_n^l$ lies in a shared image-text space, we can decompose it to a sparse set of text directions. We use a sparse coding method (Pati et al., 1993) to mine for a small set of texts for each neuron, out of a large pool of descriptions. We evaluate the found texts across different initial pools with different set sizes.

**Decomposing a neuron into a sparse set of descriptions.** Given the first principal component of the second-order effect of each neuron, $r_n^l$, we will decompose it as a sparse sum of text directions $t_j$: $r_n^l \approx \hat{r}_n^l = \sum_{j=1}^M \gamma_j^{l,n} M_{\text{text}}(t_j)$. To do this, we start from a large pool $T$ of $M$ text descriptions (e.g. the most common words in English). We apply a sparse coding algorithm to approximate $r_n^l$ as the sum above, where only $m$ of the $\gamma_j^{l,n}$'s are non-zero, for some $m \ll M$.

**Experimental settings.** We verify that the reconstructed $\hat{r}_n^l$ from the text representations captures the variation in the image representation, as measured by zero-shot accuracy on ImageNet. We simul-

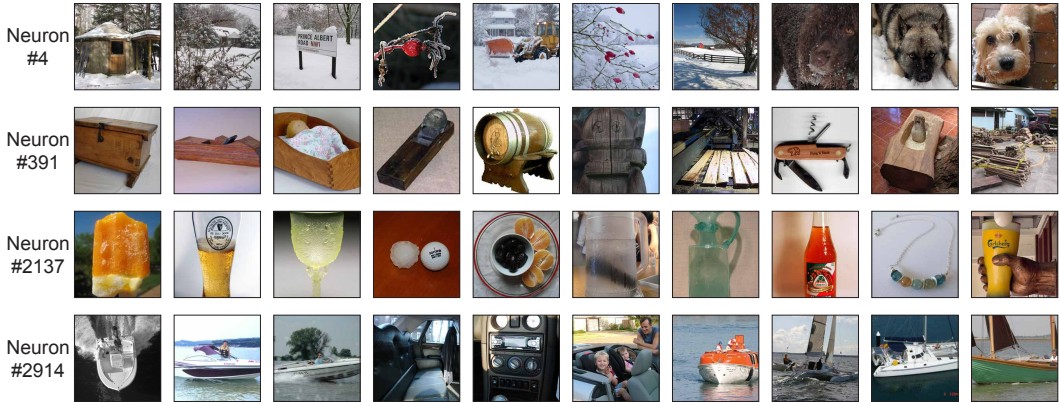

Figure 5: **Images with largest second-order effect norm per neuron.** We present the top images from 10% of ImageNet validation set for the neurons in Table 2. Note that neurons are polysemantic - they have large second-order effects on images that show multiple concepts (e.g. cars and boats). See top-50 images in Figure 13.

taneously replace the neurons' second-order contributions in a single layer with the approximation $x_n^l(I)\hat{r}_n^l + b_n^l$.

To obtain sparse decomposition for each neuron, we use scikit-learn's implementation of orthogonal matching pursuit (Pati et al., 1993). We consider two strategies for constructing the pool of text descriptions $T$. The first type is single words - the 10k and 30k most common words in English. The second type is image descriptions - we prompt ChatGPT-3.5 to produce descriptions of images that include an object of a specific class. Repeating this process for all the ImageNet (IN) classes results in ~28k unique image descriptions. We then evaluate the reconstruction of $r_n^l$ for different $m$'s and pools.

**Effect of sparse set size $m$ and different pools.** We experiment with $m \in \{4, 8, 16, 32, 64, 128\}$ and the three text pools, and present the accuracy on 10% of ImageNet validation set in Figure 4. We approach the original classification accuracy with 128 text descriptions per neuron reconstruction $\hat{r}_n^l$. Using full descriptions outperforms using single words for the text pool, but the gap vanishes for larger $m$.

**Qualitative results.** We present the images with the largest second-order norms in Figure 5, and the corresponding top-4 text descriptions in Table 2. As shown, the found descriptions match the objects in the top 10 images. Moreover, some individual neurons correspond to multiple concepts (e.g. writing both toward "yacht" and a type of a car - "cabriolet"). This property is even more apparent if more nearest neighbors are presented (see Figure 13 for the top 50 nearest neighbors). This corroborates with previous literature on neurons' polysemantic behavior (Elhage et al., 2022) - single neurons behave as a superposition of multiple interpretable features. This property will allow us to generate adversarial images in Section 5.1.

# 5 Applications

## 5.1 Automatic generation of adversarial examples

The sparse decomposition of $r_n^l$'s allows us to find overlapping concepts that neurons are writing to. We use these spurious cues to generate semantic adversarial images. Our pipeline, shown in Figure 1, mines for spurious words that correlate with the incorrect class in CLIP (e.g. "elephant", that correlates with "dog"), combines them into image descriptions that include the correct class name ("cat"), and generates adversarial images by providing these descriptions to a text-to-image model. We explain the steps in the pipeline and provide quantitative and qualitative results.

**Finding relevant spurious cues in neurons.** Given two classes $c_1$ and $c_2$, we first select neurons that contribute the most toward the classification direction $v = M_{\text{text}}(c_2) - M_{\text{text}}(c_1)$, then mine their sparse decompositions for spurious cues. Specifically, we extract the set of neurons $\mathcal{N}$ whose

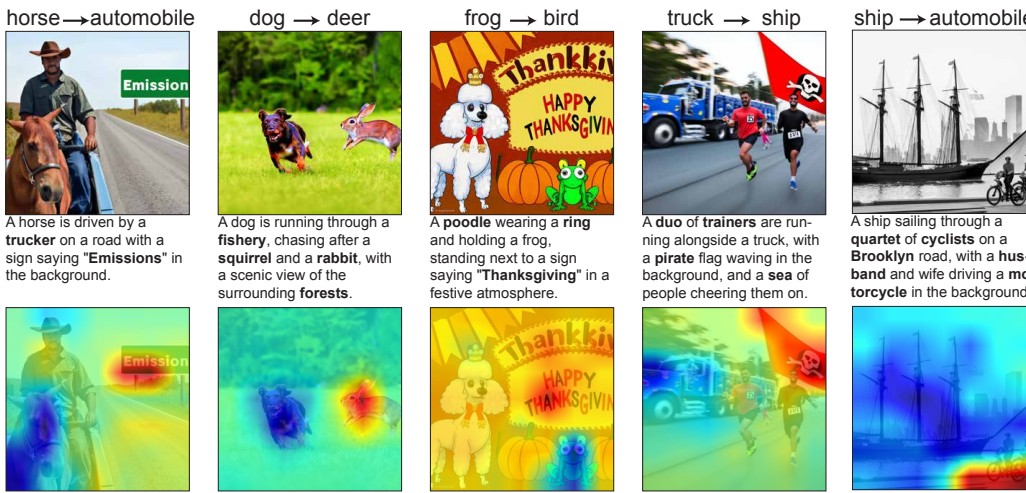

Figure 6: **Adversarial images generated by our method.** For each binary classification task, we present the generated images, the input text to the text-to-image model (words from $W^v$ are bold), and an attribution map (Gandelsman et al., 2024) for the classification (areas that contribute to the incorrect class score are red). See additional results in Figure 15.

directions are most similar to $v$: $\mathcal{N} = \text{top-k}_{n \in N} |\langle v, r_n^l \rangle|$. Utilizing the sparse decomposition from before, we compute a *contribution score* $w_j$ for each phrase $j$ in the pool $T$:

$$w_j^v = \sum_{n \in \mathcal{N}} \gamma_j^{l,n} \langle v, r_n^l \rangle. \tag{6}$$

This looks at the weight that each neuron in $\mathcal{N}$ assigns to $j$ in its sparse decomposition, weighted by how important that neuron is for classification. A phrase with a high contribution score has significant weight in one or more important neurons, and so is a potential spurious cue. The top phrases, sorted by the contribution score are collected into a set of phrase candidates $W_v$.

**Generating "semantic" adversarial examples.** We use text and image generative models to create examples with the object $c_2$ that are classified as $c_1$. First, we generate image descriptions with a large language model (LLM) by providing it phrases from the set $W^v$ and the class name $c_1$ and prompting it to generate image descriptions that include elements from both. We prompt the model to exclude anything related to $c_2$ from the descriptions and use visually distinctive words from $W_v$.

The resulting descriptions are fed into a text-to-image model to generate the adversarial images. Note that the adversarial images lie on the manifold of generated images, differently from non-semantic adversarial attacks that modify individual pixels.

**Experimental settings.** We generate adversarial images for classifying between pairs of classes from CIFAR-10 (Krizhevsky, 2009). We use the 30k most common words as our pool $T$. We choose the top 100 neurons from layers 8-10 for $\mathcal{N}$, and the top 25 words according to their contribution scores for prompting the LLM. We prompt LLaMA3 (Touvron et al., 2023) to generate 50 descriptions for each classification task (see prompt in Appendix A.7). We then filter out descriptions that include the class name and choose 10 random descriptions. We generate 10 images for each description with DeepFloyd IF text-to-image model (StabilityAI, 2023). This results in 100 images per experiment. We repeat the experiment 3 times and manually remove images that include $c_2$ objects or do not include $c_1$ objects.

We report three additional baselines. First, we repeat the same process with 100 random neurons instead of the set $\mathcal{N}$. Second, we repeat the same generation process with sparse text decompositions computed from the first principle components of the indirect effects instead of the second-order effect. Third, we do not rely on the neuron decompositions, and instead prompt the language model with the words from $M$ for which their text representations are the most similar to $v$. Both for our pipeline and the baselines, we automatically filter out synonyms of $c_2$ from the phrases provided to the language model according to their sentence similarity to $c_2$ (Reimers & Gurevych, 2019).

| Task | Random | Indirect effect | Similar words | Second order |
|---|---|---|---|---|
| horse → automobile | 1.0 (±1.4) | 2.8 (±3.7) | 1.0 (±1.4) | **5.3** (±1.9) |
| dog → deer | 0.3 (±0.5) | 6.3 (±4.8) | 3.3 (±0.9) | **22.7** (±0.5) |
| bird → frog | 0.3 (±0.5) | 1.0 (±1.4) | 5.0 (±2.9) | **8.0** (±4.5) |
| ship → truck | 0.0 (±0.0) | 0.0 (±0.0) | 0.0 (±0.0) | **5.7** (±0.9) |
| ship → automobile | 1.3 (±1.9) | 0.0 (±0.0) | 1.3 (±0.9) | **7.0** (±4.5) |

Table 3: **Accuracy of adversarial images.** We report how many generated images out of 100, fooled the binary classifier (standard deviation in parentheses).

| | Pix. Acc. ↑ | mIoU ↑ | mAP ↑ |
|---|---|---|---|
| Partial-LRP (Voita et al., 2019) | 55.0 | 35.5 | 66.9 |
| Rollout (Abnar & Zuidema, 2020) | 61.8 | 42.6 | 74.0 |
| LRP (Binder et al., 2016) | 62.9 | 35.8 | 68.5 |
| GradCAM (Selvaraju et al., 2017) | 67.3 | 39.3 | 61.9 |
| Chefer et al. (2021) | 68.9 | 49.1 | 79.7 |
| Raw-attention | 69.6 | 49.8 | 80.0 |
| TextSpan (Gandelsman et al., 2024) | 76.5 | 58.1 | 84.1 |
| Ours | **78.1** | **59.0** | **84.9** |

Table 4: **Segmentation performance on ImageNet-segmentation.** Our zero-shot segmentation is more accurate than previous methods across all metrics.

**Quantitative results.** The classification accuracy results for the adversarial images are presented in Table 3. The success rate of our adversarial images is significantly higher than the indirect effect baseline, the similar words baseline, and the random baseline, which succeeds only accidentally. For the task of generating "ship" images the will be missclassified as "truck", no other baseline manged to generate *any* adversarial images, while ours generated 5.7 images on average.

**Qualitative results.** Figure 6 includes generated adversarial examples and the descriptions that were used in their generation. The presented attribution heatmaps (Gandelsman et al., 2024) show that the found spurious objects from $W_v$ contribute the most to the misclassification, while the object from the correct class (e.g. a horse in the left-most image) contributes the least. We provide more results for additional classification tasks (e.g. "stop-sign v.s. yield") in Figure 15.

We show that understanding internal components in models can be grounded by exploiting them for adversarial attacks. Our attack is optimization-free and is not compute-intensive. Hence, it can be used for measuring interpretability techniques, with better understanding leading to improved attacks.

## 5.2 ZERO-SHOT SEGMENTATION

Finally, we use our understanding of neurons for zero-shot segmentation. Each neuron corresponds to an attribution map, by looking at its activations $p_i^{l,n}(I)$ on each image patch. Ensembling all the neurons that contribute towards a concept results in an aggregated attribution map that can be binarized to generate reliable segmentations.

Specifically, to generate a segmentation map for an image $I$, we find a set of neurons with the largest absolute value of the dot product with the encoded class name $c_i$ we aim to segment: $|\langle r_n^l, M_{\text{text}}(c_i) \rangle|$. We then average their spatial activation maps $p_i^{l,n}(I)$, standardize the average activations into $[0, 1]$, and binarize the values into foreground/background segments by applying a threshold of 0.5.

**Segmentation results.** We provide results on ImageNet-Segmentation (Guillaumin et al., 2014), which includes foreground/background segmentation maps of ImageNet objects. We use activation maps from the top 200 neurons of layers 8-10. Table 4 presents a quantitative comparison to previous explainability methods. Our method outperforms other zero-shot segmentation methods across all standard evaluation metrics. We provide qualitative results before thresholding in Figure 7. While the first-order effects ("TextSpan") highlight individual discriminative object parts, our heatmaps capture more parts of the full object.

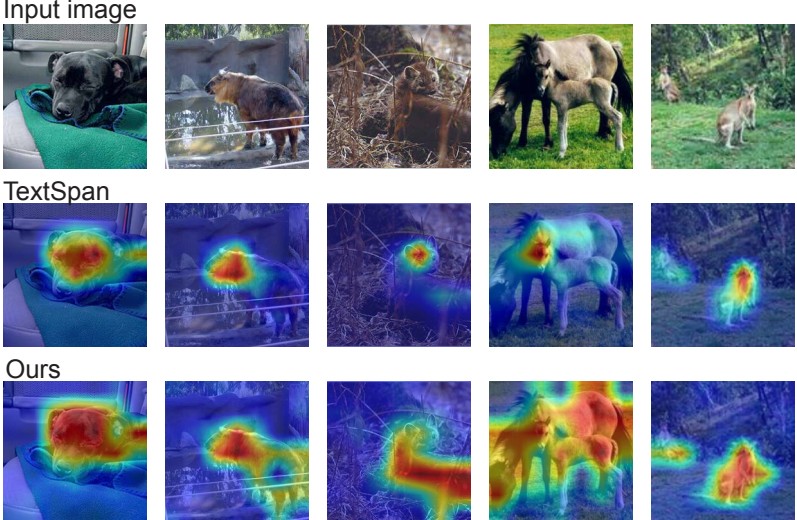

Figure 7: **Qualitative results on ImageNet-Segmentation (ViT-B-32).** Our heatmaps capture more object parts than the first-order token decomposition of Gandelsman et al. (2024).

## 6   LIMITATIONS AND DISCUSSION

We analyzed the second-order effects of neurons on the CLIP representation and used our understanding to perform zero-shot segmentation and generate adversarial images. We present mechanisms that we did not analyze in our investigation and conclude with a discussion of broader impact and future directions.

**Neuron-attention maps mechanisms.** We investigated how the neurons flow through individual consecutive attention *values*, and ignored the effect of neurons on consecutive queries and keys in the attention mechanism. Investigating these effects will allow us to find neurons that modify the attention map patterns. We leave it for future work.

**Neuron-neuron mechanisms.** We did not analyze the mutual effects between neurons in the same layer or across different layers. Returning to our adversarial "frog/bird" attack example, a neuron that writes toward "dog" may not be activated if a different neuron writes simultaneously toward "frog", thus reducing our attack efficiency. While we can still generate multiple adversarial images, we believe that understanding dependencies between neurons can improve it further.

**Future work and broader impact.** The mass production of adversarial images can be harmful to systems that rely on neural networks (e.g., the adversarial attack that causes misclassification between "yield" and "stop sign" in Figure 15). Automatic extraction of such cases allows the defender to be prepared for them and, possibly, fine-tune the model on the generated images to avoid such attacks. We plan to investigate this approach to improve CLIP's robustness in future work.

Currently, our attack pipeline relies on a few independent components, each of which has failure modes. For example, the language model can fail to generate a coherent sentence that includes *many* phrases from $W_v$, and can omit the class name $c_2$ or accidentally include the class name $c_1$. Additionally, the text-to-image model can fail to generate an image that follows the exact description and can drop crucial elements from the description. We believe that future improvements in the language and vision models will increase the success rate of our attack, and plan to continue to develop and improve it in the future.

**Acknowledgments.** We would like to thank Alexander Pan for helpful feedback on the manuscript. YG is supported by the Google Fellowship. AE is supported in part by DoD, including DARPA's MCS and ONR MURI, as well as funding from SAP. JS is supported by the NSF Awards No. 1804794 & 2031899.

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

# A APPENDIX

## A.1 SECOND ORDER ABLATIONS FOR VIT-L

We repeat the same experiments from Section 3.3 for ViT-L-14, trained on LAION dataset (Schuhmann et al., 2022). For this model, we only use 10% of ImageNet validation set. Here, the maximal drop in performance when ablating the second order is relatively smaller and is spread across more layers. Nevertheless, the same properties presented and discussed in Section 3.3 hold for this model.

## A.2 FIRST ORDER ABLATIONS

For the two models discussed above, ViT-B-32 and ViT-L-14, we provide the mean-ablation results for the first-order effects of MSA layers, as computed in Gandelsman et al. (2024). For each model, we present the performance before and after accumulative mean-ablation of all the first-order effects of MSA layers. As shown in Figure 11 and Figure 12, the neurons with the significant second-order effects appear right before the layers with the significant first-order effects.

## A.3 ADDITIONAL ADVERSARIAL IMAGES

We present additional semantic adversarial results, generated by our method for ViT-B-32, in Figure 15. We demonstrate a wide variety of tasks, including additional pairs from CIFAR-10 dataset, and adversarial attacks related to traffic signs (e.g. misclassification between a stop sign and a yield sign or a crossroad). For each image, we provide the text used for generating it, and highlight the spurious cues words from the sparse decompositions.

## A.4 ADDITIONAL SPARSE DECOMPOSITION RESULTS

We provide additional examples of sparse decompositions of neurons in Table 5 and the images with the top norms for the second-order effects of the same neurons in Figure 14. As shown, the found descriptions match the objects in the top 10 images.

## A.5 CONCEPT DISCOVERY IN IMAGES

We present an additional application - concept discovery in images. We aim to discover concepts in image $I$, by aggregating phrases that correspond to the neurons that are activated on $I$. Here, we start from the set of *activated* neurons $\mathcal{N}$ (for which $||\phi_n^l(I)||_2$ is above the 98th percentile of norms computed across ImageNet images). Similarly to the contribution score described in Section 5.1, we compute an *image-contribution score* $w_j^I$ for each phrase $j$ according to its combined weight in the decompositions of neurons in $\mathcal{N}$. Formally, $w_j^I$ is the overall sum of weights that each neuron in $\mathcal{N}$ assigns to $j$ in its decomposition, weighted by the neuron second-order norms: $w_j^I = \sum_{n \in \mathcal{N}} \gamma_j^{l,n} ||\phi_n^l(I)||_2$. The phrases with the highest image-contribution score are picked to describe the image concepts.

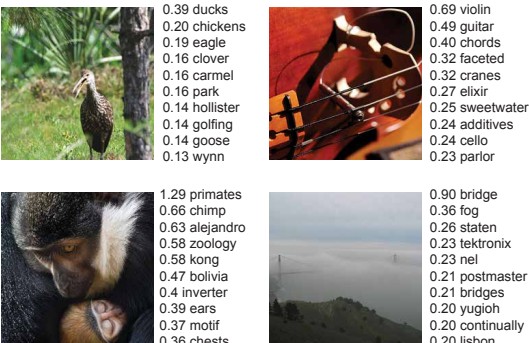

Figure 8: **Concept discovery in images (ViT-B-32).** We include top-10 words discovered by aggregating words in sparse decompositions of activated neurons.

**Qualitative results.** We present qualitative results for neurons and the top-10 discovered concepts from layer 9 of ViT-B-32 in Figure 8, when using the most common words as the pool. The number of neurons activated on these images, $|\mathcal{N}|$, is between 29 and 59, less than 2% of the neurons in the layer. Nevertheless, the top words extracted from these neurons relate semantically to the objects in the image and their locations. Surprisingly, the top word for each of the images appears only in one or two of the neuron sparse decompositions and is not spread across many activated neurons.

We acknowledge that while this application discovers meaningful concepts that correspond to the input images, there are other approaches for extract these concepts (e.g. sparsely decomposing the image representation, as shown in Bhalla et al. (2024)).

## A.6 Derivations with Layer Normalization

In many implementations of CLIP, there is a layer normalization between the Vision Transformer and the projection layer $P$. In this case, the representation is:

$$M_{\text{image}}(I) = P(LN(\text{ViT}(I))) \tag{7}$$

where the $LN$ is the layer normalization. Specifically, $LN$ can be written as:

$$LN(x) = \gamma * \frac{x - \mu}{\sqrt{\sigma^2 + \epsilon}} + \beta = \underbrace{\left[\frac{\gamma}{\sqrt{\sigma^2 + \epsilon}}\right]}_{=A} * x - \underbrace{\left[\frac{\mu\gamma}{\sqrt{\sigma^2 + \epsilon}} - \beta\right]}_{=B}, \tag{8}$$

where $x \in \mathbb{R}^{d'}$ is the input token, $\mu_l, \sigma_l \in \mathbb{R}$ are the mean and standard deviation, and $\gamma, \beta \in \mathbb{R}^{d'}$ are learned vectors. To include $A$ and $B$ in the second-order effect of a neuron flow, we replace the input-independent component in Equation (5), $PW_{VO}^{l',h}w^{l',n}$, with:

$$P(A * W_{VO}^{l',h}w^{l,n} + \frac{B}{c}) \tag{9}$$

Where $c$ is a normalization constant that splits $B$ equally across all the neurons that can additively contribute to it.

Except for the layer normalization before the projection layer, the input to the MSA layers that comes from the residual stream also flows through a layer normalization. Thus, if the input to the MSA layer in layer $l$ is the list of tokens $[z_0^l, ... z_K^l]$, the output that corresponds to the class token is:

$$\left[\text{MSA}^l([z_0, ..., z_K])\right]_0 = \sum_{h=1}^{H} \sum_{i=0}^{K} a_i^{l,h}(I) W_{VO}^{l,h} LN^l(z_i), \tag{10}$$

where $LN^l$ is the normalization layer at layer $l$, that can be parameterized similarly to Equation (8) by $A^l, B^l \in \mathbb{R}^{d'}$. We modify the definition of the second-order effect accordingly:

$$\phi_n^l(I) = \sum_{l'=l+1}^{L} \sum_{h=1}^{H} \sum_{i=0}^{K} \left(p_i^{l,n}(I) a_i^{l',h}(I)\right) \left(P\left(A * W_{VO}^{l',h}(A^l * w^{l,n} + \frac{B^{l'}}{c^{l'}}) + \frac{B}{c}\right)\right), \tag{11}$$

where $c^{l'}$ is is a normalization coefficient that splits $B^{l'}$ equally across all the neurons before layer $l'$.

In all of our experiments, we use this modification. Most of the elements in the modification add constant biases. Therefore, they can be ignored in our experiments as in many of the experiments constant biases do not change the results. For example, in our mean-ablation experiment, we subtract the mean, computed across a dataset.

## A.7 Prompts

We provide the prompt that was used for generating sentences given the set of words $W_v$, as presented in Section 5.1, in Table 6. This prompt is given to LLAMA3 model (Touvron et al., 2023).

Additionally, we provide the prompt that was used for generating the pool of ImageNet class descriptions, presented in Section 4. We prompt ChatGPT (GPT 3.5) with the prompt template provided in Table 7.

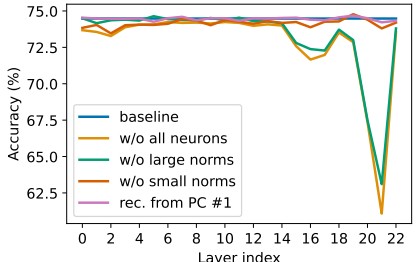 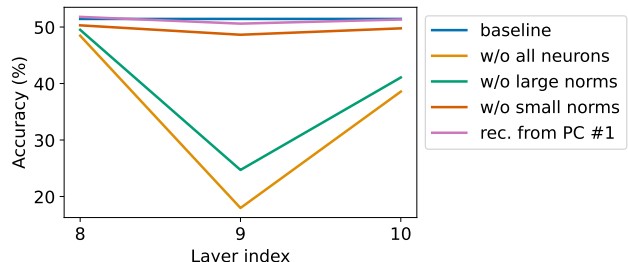

Figure 9: **ViT-L-14 second-order ablations.**

Figure 10: **Mean-ablation of second order effects on ImageNet-R (ViT-B-32, layers 8-10).** We repeat the evaluation in Figure 3 on ImageNet-R. The performance of different ablations follows the same trends as that of ImageNet.

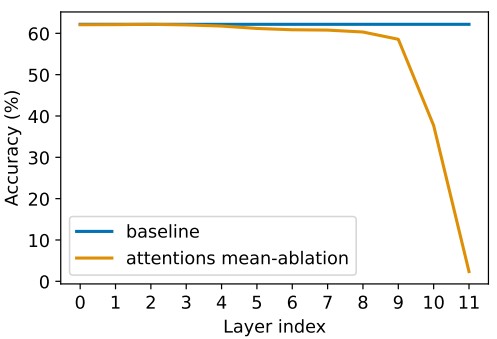 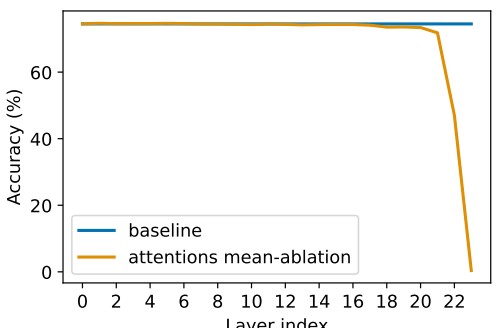

Figure 11: **ViT-B-32 first-order MSAs ablation.**  Figure 12: **ViT-L-14 first-order MSAs ablation.**

| Neuron | ImageNet class descriptions | Common words (30k) |
|---|---|---|
| #600 | +"Image with a wiry, weather-resistant coat"
+"Image showcasing a compact and lightweight sleeping bag"
+"Picture of a camper towing bicycles"
+"Image with a Border Terrier jumping" | +"tents"
+"svalbard"
+"miles"
-"mountainous" |
| #974 | -"Photograph taken during a race"
-"Silhouette of a running dog"
-"Picture taken in a fishing competition"
+"Silhouette of hammerhead shark with other ocean creatures" | +"runners"
+"races"
-"dolphin"
+"expiration" |
| #1517 | +"Chair with a foot pedal control"
-"Picture that captures the breed's intelligence"
-"Image with snow-capped mountains as scenery"
+"Image with graffiti on a train" | +"bus"
-"filings"
-"percussion"
+"wheelchairs" |
| #2002 | +"Image depicting a sustainable living option"
+"Photo taken in a train yard"
-"Image featuring snow-covered rooftops"
+"Rescue equipment" | -"genres"
+"governance"
+"gravel"
+"conserve" |

Table 5: **Additional examples of sparse decomposition results.** For each neuron, we present the top-4 texts corresponding to the sparse decomposition with $m = 128$ and the signs of the coefficients in the decomposition.

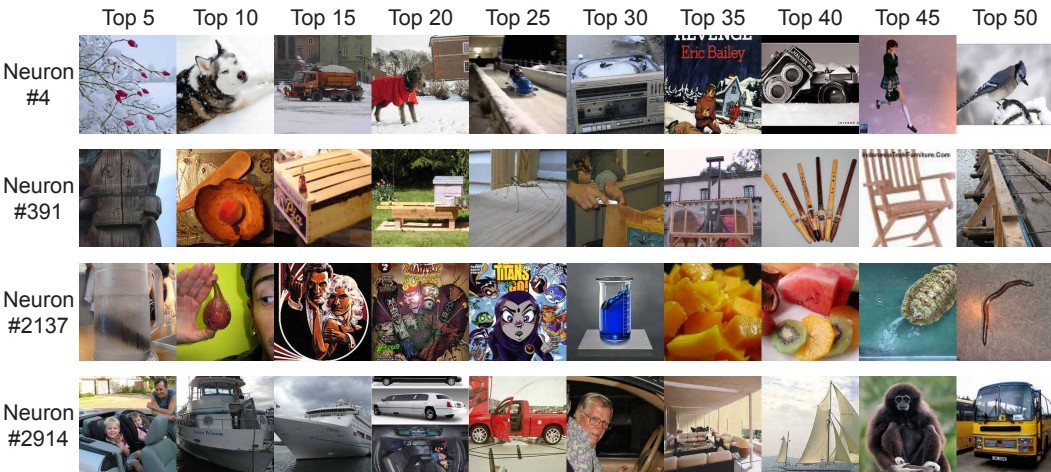

Figure 13: **Images with largest second-order effect norm per neuron.** We present the top images from 10% of ImageNet validation set, for the neurons in Table 2. Notice that additional concepts that are not captured by the top-4 descriptions in Table 2 are starting to appear.

## A.8 COMPUTE

As our method does not require additional training, the time of our experiments depends linearly on the inference time of CLIP (and other generative models that were used for the adversarial images generation), and on the number of images we use for the experiments (~5000 in our case). All our experiments were run on one A100 GPU. The most time-consuming experiment—computing the per-layer mean-ablation results for ViT-L-14—took 5 days.

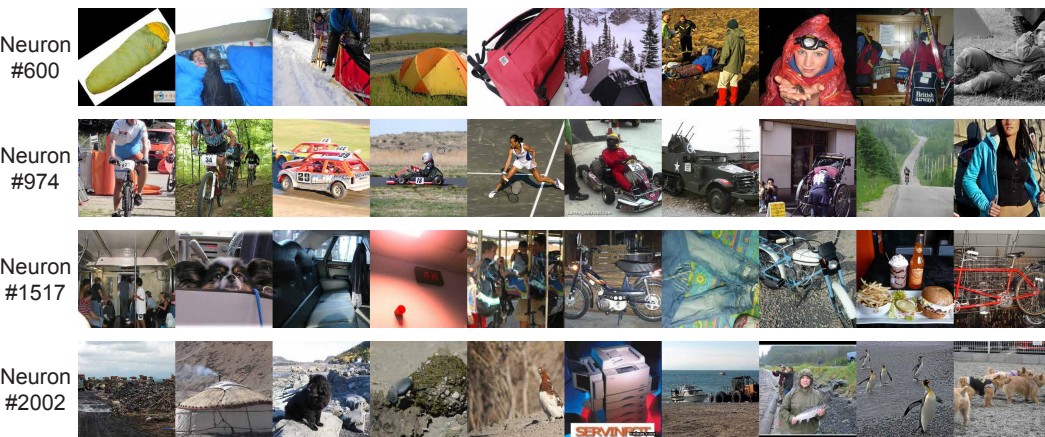

Figure 14: **Images with largest second-order effect norm per neuron.** We present the top images from 10% of ImageNet validation set, for the neurons in Table 5.

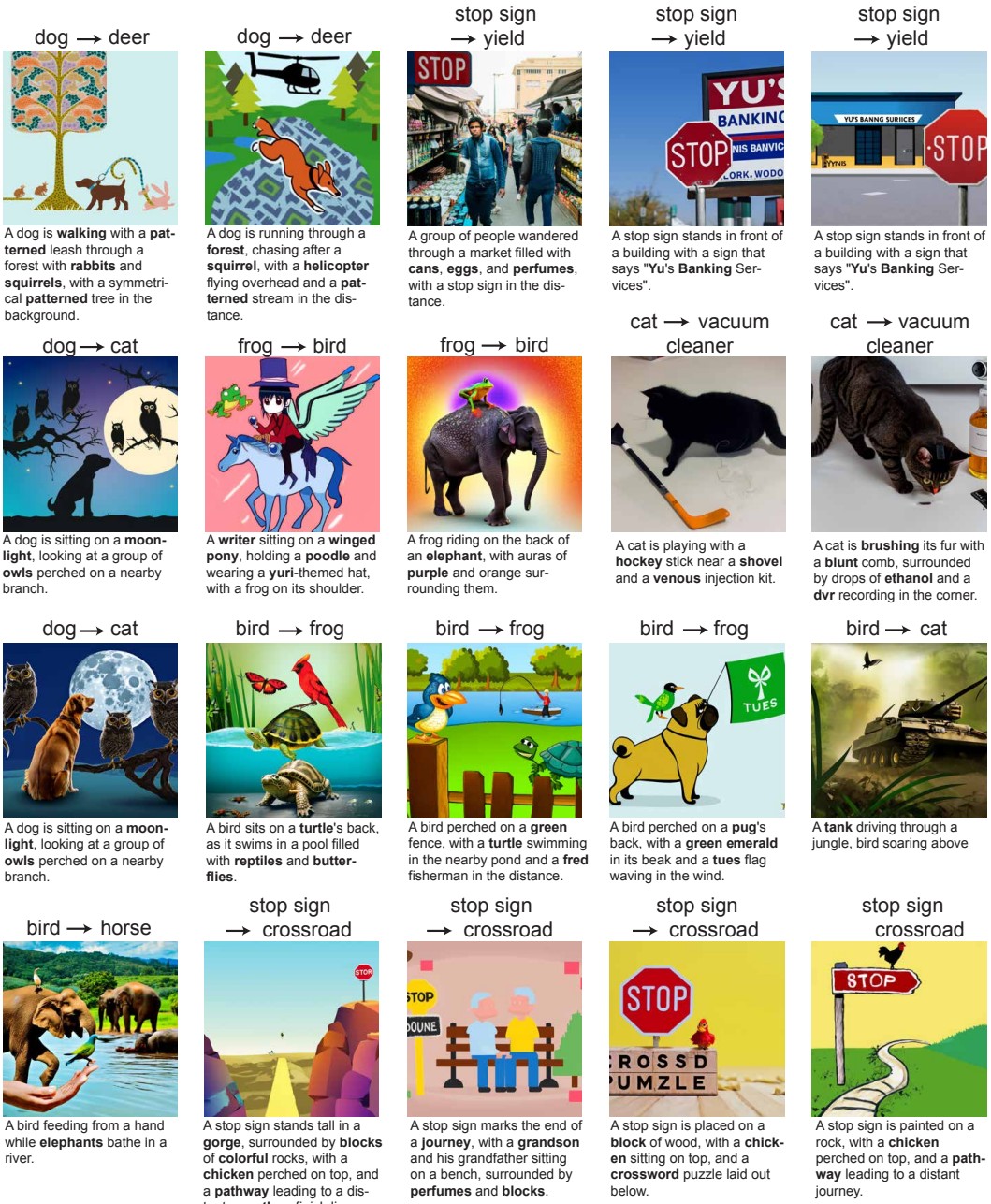

Figure 15: **Additional adversarial examples generated by our method.** We provide the sentence that was given to the text-to-image model to generate it. Words from $W^v$ are highlighted in bold.

You are a capable instruction-following AI agent.
I want to generate an image by providing image descriptions as input to a text-to-image model.
The image descriptions should be short. Each of them must include the word "{*class_1*}".
They must not include the word "{*class_2*}", any synonym of it, or a plural version!
The image descriptions should include as many words as possible from the next list and almost no other words:
{*list*}
Do not use names of people or places from the list unless they are famous and there is something visually distinctive about them. In each of the image descriptions mention as many objects and animals as possible from the list above. If you want to mention the place in which the image is taken or a name of a person, describe it with visually distinctive words. For example, if "Paris" is in the list, instead of saying "... in Paris", say "... with the Eiffel Tower in the background" or "... next to a sign saying 'Paris'". Don't mention words that are too similar to "{*class_2*}", even if they are in the list above. For example, if the word was "tree" you should not mention "trees", "bush" or "eucalyptus". Only use words that you know what they mean.
Generate a list of 50 image descriptions.

Table 6: **The language model prompt for generating image descriptions.**

Provide 40 image characteristics that are true for almost all the images of {*class*}. Be as general as possible and give short descriptions presenting one characteristic at a time that can describe almost all the possible images of this category. Don't mention the category name itself (which is "{*class*}"). Here are some possible phrases: "Image with texture of ...", "Picture taken in the geographical location of...", "Photo that is taken outdoors", "Caricature with text", "Image with the artistic style of...", "Image with one/two/three objects", "Illustration with the color palette ...", "Photo taken from above/below", "Photograph taken during ... season". Just give the short titles, don't explain why, and don't combine two different concepts (with "or" or "and").

Table 7: **The prompt for generating the pool of class descriptions.** We prompt the model with all the ImageNet classes.

