# OpenReview forum: "Interpreting the Second-Order Effects of Neurons in CLIP"
_ICLR.cc/2025/Conference — ICLR 2025 Poster_

### Official Review · Reviewer_XFDa · 2024-10-30

**Soundness:** 3
**Presentation:** 3
**Contribution:** 3
**Rating:** 8
**Confidence:** 3

**Summary:**

This paper proposes to interpret the functional of individual neurons in CLIP by investigating their "second order" effects at the joint text-image output space. Specifically, a neuron's contribution to the output feature space is defined as the summing of the effects flowing from the neuron through the later attention heads, to the output. Experimental results show several observations: (1) the effects of an individual neuron are highly selective. (2) Each neuron's contribution in the output feature space can be approximated by a single direction, which can be decomposed into representations of a sparse set of texts. (3) By exploiting the polysemantic property of neuron, it is possible to generate adversarial images that fool the CLIP classifier. (4) The semantic-awareness of neurons can be utilized for semantic segmentation.

**Strengths:**

1. The paper is well motivated, as exploring the functions of neurons in CLIP model facilitates better understanding of its representation learning mechanism, and guide further improvements
2. The paper proposes to study the "second order" effect of neurons, which is demonstrated to show a clearer signal of their contribution comparing to first order and indirect effects
3. The paper shows an interesting finding that each neuron's contribution to the joint text-image output space is approximately a single direction vector, and implies a ploysemy property.
4. The paper further exploits the finding of 3 for generating adversarial images and application of semantic segmentation, showing that by taking advantage of polysemy neuron's output representations, it's possible to fool the CLIP classifier by using words that's spuriously correlated with a wrong class; and that by aggregating the activation maps of neurons that highly correlated with a query concept, its semantic segmentation mask can be acquired.

**Weaknesses:**

Please see questions

**Questions:**

1. In the related work section, the paper lists several other neurons interpretability works that explore what concepts the neuron activated on, and saying that in contrast their work focuses on the neuron's contribution to the output space. My question is: what is the relation between the two approaches (exploring what examples the neurons activate on vs. exploring their contribution in the output space), what are the advantages / limitations of one over the other, does one show complementary aspects that the other cannot  observe?

---

> ### Author Response · Authors · 2024-11-14
>
> We thank the reviewer for their question.
>
> __“What is the relation between exploring what examples the neurons activate on vs. exploring their contribution in the output space, what are the advantages/limitations of one over the other, does one show complementary aspects that the other cannot observe?”__
>
> There are non-trivial relations between the highly activated images of a neuron (its “input space”) and its output space. Here, for example, we rely on the assumption that if a neuron’s second effect contributes toward some direction (e.g. the “dog” text representation), it probably means that the neuron is activated on images that have the concept in them. While it is mostly the case and allows us to attack the model, we found some instances where a neuron has relatively high activation patterns on images of a concept, but since the neuron is followed by a GeLU (which can produce negative values for positive inputs), its second-order contribution direction is negative. Nevertheless, in most cases, there is a clear positive correspondence between the concepts in highly activated images and the neurons’ output space. Moreover, in most cases, there is a clear correspondence between the activation patterns and the output space (e.g., a dog segment will be highly activated for a neuron that has a high contribution towards the “dog” text representation). This property allows us to mine for the neurons with a high contribution toward a text representation and use their activation maps for zero-shot segmentation (see Figure 7 and Table 4).

---

> > ### Comment · Reviewer_XFDa · 2024-11-21
> >
> > Thank you, the answer addressed my question.

---

### Official Review · Reviewer_XYxA · 2024-11-03

**Soundness:** 3
**Presentation:** 3
**Contribution:** 4
**Rating:** 8
**Confidence:** 4

**Summary:**

This paper introduces an approach to generalize the first order effects of a neuron in attention networks into second order effects through successive layers. This methodology helps uncover the sparseness of the neuron second order effect as well as the decomposition of each neurons second order effect into a single direction vector.

**Strengths:**

- Extensive empirical validation of second order effects (e.g. second order effect neuron sparseness)
- Intuitive and interesting applications of second order effect control in the semantic adversarial example generation
- Increased understanding of internal attention model mechanism through semantic adversarial examples
- Improved segmentation results over TextSpan

**Weaknesses:**

- Sparse coding to find textual descriptions of neurons may be very computationally expensive
- Not considering nonlinearities in second order effects (Eqn 5)

**Questions:**

- Are there any gradient based approaches to finding textual descriptions of each neuron's second order effect?
- Were any additional text to image models evaluated other than DeepFloyd IF, if so was there similar success to images generated with DeepFloyd IF?

---

> ### Author Response · Authors · 2024-11-14
>
> We thank the reviewer for the valuable comments and questions.
>
> __“Sparse coding to find textual descriptions of neurons may be very computationally expensive”__
>
> Empirically, we didn’t find the sparse coding technique being too expensive. We used an off-the-shelf sklearn library (CPU-based solution) and did not try to optimize this part. Applying this sparse coding technique to all the neurons we analyzed (one layer in the model) with a dictionary of ~30000 entries took a few hours (please see Appendix A.8 for more details).
>
> __“Not considering nonlinearities in second-order effects (Eqn 5)”__
>
> Indeed, for simplicity, we do not mention the nonlinearities (that come from the layer normalizations) in the main text. Nevertheless, in all of our experiments, we include this part in the computation of second-order effects. We included the extended derivations that we use to compute the second-order effects with layer normalization in Appendix A.6.
>
> __“Are there any gradient-based approaches to finding textual descriptions of each neuron's second-order effect?”__
>
> To the best of our knowledge, there are no gradient-based approaches for finding textual descriptions for neurons, mostly due to the difficulty of back-propagating into a discrete space.
>
> __“Were any additional text-to-image models evaluated other than DeepFloyd IF, if so was there similar success to images generated with DeepFloyd IF?”__
>
> We specifically used DeepFloyd IF as our model, because it does not use CLIP model in its conditioning. Nevertheless, the same approach can be applied to other generative models as well. Moreover, we suspect that with better models, that follow the prompt better, our adversarial results will improve, and we plan to analyze it in the future.

---

> > ### Author Response · Authors · 2024-11-23
> >
> > Dear Reviewer,
> >
> > Thank you for taking the time to review our paper and provide valuable feedback. As the discussion phase is nearing its conclusion and there will be no second stage of author-reviewer interactions, we would like to confirm if our responses from a few days ago have effectively addressed your concerns. We hope they have resolved the issues you raised. If you require further clarification or have additional questions, please don’t hesitate to reach out. We are happy to continue the conversation.
> >
> > Thank you,
> >
> > The authors

---

### Official Review · Reviewer_M6ey · 2024-11-03

**Soundness:** 2
**Presentation:** 2
**Contribution:** 2
**Rating:** 6
**Confidence:** 3

**Summary:**

The paper presents an interpretability study focused on understanding the second-order effects of neurons in CLIP. The authors propose a novel "second-order lens" to analyze neuron contributions that flow through attention heads to the model output.

**Strengths:**

1. The technical contributions are sound  and interesting.
2. The paper is well written.
3. The paper included thorough evaluations.

**Weaknesses:**

Generally good paper so please see questions.

**Questions:**

1. What happens if you apply the same method to the text encoder in CLIP?
2. Have the authors tried it on other variants of CLIP like MaskCLIP?
3. Do the findings of this paper apply to other domains like medical imaging (MedCLIP)?

---

> ### Author Response · Authors · 2024-11-14
>
> We thank the reviewer for the valuable questions.
>
> __“What happens if you apply the same method to the text encoder in CLIP?”__
>
> Although the same technique can be applied to the text encoder, we did not do that. More specifically, the text encoder is also a transformer, so we can compute the second-order effects similarly, and then apply sparse decomposition into a dictionary of text/image representations. We focused here on the vision encoder for exploring the polysemantic behavior of vision neurons with text and to demonstrate that the second order is useful for finding adversarial images. We plan to explore the text encoder in future work.
>
> __“Have the authors tried it on other variants of CLIP?”__
>
> We evaluated our model on different model sizes and found similar trends (please see Figures 9 and 12 in the supplementary material).
>
> __“Do the findings of this paper apply to other domains like medical imaging (MedCLIP)?”__
>
> We did not apply it to medical imaging but we think that the same findings can be applied there as well, as the architecture of the model is similar.

---

> > ### Author Response · Authors · 2024-11-23
> >
> > Dear Reviewer,
> >
> > Thank you for taking the time to review our paper and provide valuable feedback. As the discussion phase is nearing its conclusion and there will be no second stage of author-reviewer interactions, we would like to confirm if our responses from a few days ago have effectively addressed your concerns. We hope they have resolved the issues you raised. If you require further clarification or have additional questions, please don’t hesitate to reach out. We are happy to continue the conversation.
> >
> > Thank you,
> >
> > The authors

---

> > > ### Author Response · Authors · 2024-11-27
> > >
> > > Dear Reviewer,
> > > Thank you again for your valuable comments. We would like to ask once again if our rebuttal addressed your concerns and if there is anything else that can resolve the issues you raised.
> > >
> > > Thank you,
> > >
> > > The authors

---

### Official Review · Reviewer_g9ZC · 2024-11-04

**Soundness:** 3
**Presentation:** 2
**Contribution:** 2
**Rating:** 5
**Confidence:** 4

**Summary:**

This work presents a novel approach for examining potential second-order effects of neurons of CLIP representations and how these can be used in the context of zero-shot segmentation and generation of adversarial images.

To this end, the authors focus on the contribution of each individual neuron to the output in terms of second-order effects via the computation of the additive contribution of each neuron through the MSAs and projection to the input space.

The experimental evaluation focuses on the empirical analysis of the obtained effects and how these insights can be used in the context of generating "semantic" adversarial examples and using said effects for zero-shot segmentation.

**Strengths:**

This paper draws inspiration from recent approaches that aim to examine and evaluate the functionality of each neuron in a given architecture. Automated interpretability constitutes an important challenge for modern architectures and this work aims to approach this in a different way via the contribution of neurons to the output representation and the information flow through the MSA blocks.

**Weaknesses:**

The connection of the proposed approach to highly relevant work is a bit lacking. Can the authors provide a discussion on [1], highlighting the differences in the decomposition and analysis of the direct effects of the neurons?

I find the focus on a single dataset, i.e., ImageNet, to be a bit restrictive in terms of analysing the behavior of the proposed approach. Indeed, most approaches in this line of work considered additional datasets, e.g., Waterbirds, CUB and Places365. The same applies for the adversarial examples setting, where the authors only consider CIFAR-10. What happens when trying to generate adversarial examples in a more complicated dataset, e.g., CUB or ImageNet?

Did the authors reproduce the results for all the method in Table 4? Since they are different than the ones reported [1], I would expect that to be the case.

Can the authors provide the weights of the texts corresponding to the sparse decomposition of each neuron? A full list for some neurons would also be helpful.
Qualitatively, what are the differences when choosing a different sparse set size m.

What is the motivation behind the consideration a binary classification task instead of a classical setting? What classifier is used and how is it trained?

The authors mention that the generated adversarial images lie on the manifold of generated images differently from non-semantic adversarial attacks. Can the authors elaborate on this claim?

What is the complexity of the proposed approach compared to other interpretability focused methods?

[1] Gandelsman et al.,  Interpreting CLIP’s image representation via text-based decomposition, ICLR 2024

**Questions:**

Please see the Weaknesses section.

---

> ### Author Response · Authors · 2024-11-14
>
> We thank the reviewer for the valuable comments and questions.
>
> __“Can the authors provide a discussion on TextSPAN, highlighting the differences in the decomposition and analysis of the direct effects of the neurons?”__
>
> [1] explores the direct effect of different components in CLIP. As shown in [1], the direct effects of the MLPs are very low – mean-ablating the direct effects of all the neurons in all the layers results in a negligible drop in performance (please see Table 1 in [1]). Moreover, the main observation is that most of the direct effect comes from the late attention layers. Therefore, in our analysis, we look at the information flow from neurons through such attention layers, directly to the output. A different way of looking at it is that in our analysis we decompose the contributions of attention layers into further flows from previous layers through them, and focus on the flows from individual neurons.
>
> __“The focus on a single dataset, i.e., ImageNet, is a bit restrictive in terms of analyzing the behavior of the proposed approach”__
>
> Indeed, we mostly focus on ImageNet classification as our downstream evaluation metric, to analyze how much our ablations change the final representations, similarly to [1]. Nevertheless, as shown in Figure 10, performing similar analyses on other datasets (e.g., ImageNet-R) shows similar trends.
>
> __“What happens when trying to generate adversarial examples in a more complicated dataset?”__
>
> Our approach for adversarial attacks does not rely on the images of the datasets. We only use the class names from the datasets to showcase that we can generate an image that will be classified as one class while having objects from the other class. We specifically used classes that look very different (random CIFAR class pairs - e.g., horse and automobile) to produce those examples. We provide additional examples for other classes in Figure 15.
>
> __“Did the authors reproduce the results for all the methods in Table 4?”__
>
> Yes, we reproduced all the baselines and re-evaluated the different approaches. The difference in numbers from TextSpan is due to a different choice of a model – we focus on the model that was pre-trained by OpenAI on their data, while TextSpan focuses on a model that was trained by OpenCLIP on LAION.
>
> __“Qualitatively, what are the differences when choosing a different sparse set size m?”__
>
> We found that different m’s provide relatively similar results when looking at the top coefficients. Nevertheless, larger m’s allow better reconstruction of the second-order of a neuron. That suggests that neurons are polysemantic and correspond to multiple texts, but many of them have only a few significant and coherent text directions for which they are responsible.
>
> __“What is the motivation behind the consideration of a binary classification task instead of a classical setting? What classifier is used and how is it trained?”__
>
> We aim to produce an adversarial attack on CLIP. Therefore, we use CLIP as the zero-shot classifier, by comparing the output image representation to the text representations of the two class descriptions (e.g., “An image of a dog”/”An image of a cat”).
>
> Typically, adversarial attacks are images that cause a classifier to label them as one class while they are perceived by humans as another class. This is the reason for choosing pairs of classes.  As there is no training or any other usage of the dataset except for the class names, we decided to choose class pairs that are very different and CLIP can usually easily distinguish between them (e.g., ship and truck). We show additional examples from other pairs of classes (not only from CIFAR10) in Figure 15.
>
> __“The authors mention that the generated adversarial images lie on the manifold of generated images differently from non-semantic adversarial attacks. Can the authors elaborate on this claim?”__
>
> Classical (non-semantic) adversarial attacks modify images directly by adding pixel noise to the images, which causes the image to be classified incorrectly. Differently from this approach, our images are generated from a generative model that is trained to model the natural image distribution (for example – a diffusion model). We do not tamper with the generative model or the image pixels to make it adversarial.

---

> > ### Author Response · Authors · 2024-11-14
> > **cont.**
> >
> > __“What is the complexity of the proposed approach compared to other interpretability-focused methods?”__
> >
> > Current interpretability approaches range from simple decomposition-based approaches (e.g. modeling the direct effect of components - TextSpan [1]/LogitLens [2]) to training-based approaches that aim to learn an interpretable decomposition of the representation (e.g. Sparse Autoencoders [3]). Our method does not rely on learning and instead uses classical sparse decomposition techniques, which makes it less compute-intensive. On the other hand, it looks at more specific computational paths than the direct effects or the indirect effect, and therefore it can provide in some cases more useful interpretation that can be used for improving downstream performance (see Table 3). As shown in A.8, the compute recruitments of this method are not significant in comparison to the training-based approaches.
> >
> > [2] Nostalgebraist, interpreting GPT: the logit lens, AI Alignment Forum 2022
> >
> > [3] Cunningham et al., Sparse Autoencoders Find Highly Interpretable Features in Language Models, ICLR 2024
> >
> >
> > __“Can the authors provide the texts' weights corresponding to each neuron's sparse decomposition? A full list of some neurons would also be helpful.”__
> >
> > We provide an example for all the 128 text representations that correspond to neuron 4, layer 9 in the model and their corresponding weight coefficients (see Figure 5 for corresponding images with a highest second-order effect):
> >
> > | Text         | Value | Text       | Value | Text     | Value  | Text   | Value |
> > |--------------|-------------|--------------|-------------|--------------|-------------|--------------|-------------|
> > | snowy | 0.28    | fuchs | -0.07   | discus | -0.05   | plato | 0.03    |
> > | frost      | 0.19    | timberland | 0.07    | treasury | -0.05   | transistor       | 0.03    |
> > | closings | 0.16    | thunderstorm     | -0.07   | narnia | 0.05    | viper | -0.03   |
> > | advent   | 0.15    | nav | -0.07   | tasmanian        | 0.05    | sligo | 0.03    |
> > | southwark        | -0.14   | auditorium       | -0.07   | burnley | 0.05    | pendleton        | -0.03   |
> > | grassy | -0.13   | chinatown        | -0.07   | elk | 0.05    | purdue | 0.03    |
> > | subcontractors   | -0.12   | atherosclerosis  | 0.07    | mosque | 0.05    | loss | -0.03   |
> > | wipers  | 0.11    | soybeans         | 0.07    | chores | 0.05    | bild | 0.03    |
> > | smugmug | 0.11    | ridges | -0.06   | couplings        | -0.05   | composites       | -0.03   |
> > | molten | 0.11    | naxos | -0.06   | kemp | -0.05   | packard | -0.03   |
> > | bosses | -0.11   | kazakhstan       | 0.06    | chinook | 0.05    | slr | -0.03   |
> > | angola | -0.11   | macedonian       | -0.06   | charleston       | -0.05   | steamboat        | 0.03    |
> > | reflective       | -0.11   | autoconf         | -0.06   | chocolate        | 0.05    | lesotho | 0.02    |
> > | ozark | -0.11   | dispenser        | -0.06   | woodstock        | -0.05   | genie | -0.02   |
> > | grouping         | 0.1     | pupils | -0.06   | pcos | 0.05    | griffith         | -0.02   |
> > | oakville         | 0.1     | firenze | 0.06    | dhl | -0.04   | garrison         | -0.02   |
> > | mikasa | -0.1    | ledger | -0.06   | manning | -0.04   | telluride        | -0.02   |
> > | exhibitionism    | -0.1    | beckham | 0.06    | amtrak | 0.04    | regency | 0.02    |
> > | horns | -0.1    | sasha | 0.06    | swimsuit         | -0.04   | xtc | 0.02    |
> > | anyone | -0.09   | pistons | 0.06    | peril | 0.04    | platelet         | -0.02   |
> > | orbit | -0.09   | kauai | 0.06    | hopper | 0.04    | coldplay         | -0.02   |
> > | testimonial      | -0.09   | banded | 0.06    | chairs | 0.04    | gump | -0.02   |
> > | clemente         | -0.09   | susceptible      | 0.06    | forte | 0.04    | shakira | -0.02   |
> > | orbit        | -0.09   | kauai        | 0.06    | hopper       | 0.04    | coldplay     | -0.02   |
> > | testimonial  | -0.09   | banded       | 0.06    | chairs       | 0.04    | gump         | -0.02   |
> > | clemente     | -0.09   | susceptible  | 0.06    | forte        | 0.04    | shakira      | -0.02   |
> > | vail   | 0.09    | tolkien      | -0.06   | bonnet | 0.04    | atpase | 0.02    |
> > | mimic | -0.09   | tortoise     | -0.06   | prominence   | 0.04    | weir | 0.02    |
> > | clay | -0.08   | braille      | -0.06   | drosophila   | 0.04    | lama | -0.02   |
> > | sprite       | -0.08   | counters     | 0.05    | neurosurgery | -0.04   | greyhound    | -0.02   |
> > | supervised   | 0.08    | september    | -0.05   | brackets     | 0.04    | clerks       | 0.02    |
> > | hardwood     | -0.08   | martial      | 0.05    | srv | 0.03    | social       | 0.01    |
> > | oj | 0.08    | flaws        | -0.05   | visor        | -0.03   | monet        | 0.01    |
> > | celsius      | 0.08    | airbrush     | -0.05   | commodore    | 0.03    | restaurant   | 0.01    |
> > | sidewalk     | 0.07    | sfo | -0.05   | maintained   | -0.03   | peridot      | 0.01    |

---

> > > ### Author Response · Authors · 2024-11-23
> > >
> > > Dear Reviewer,
> > >
> > > Thank you for taking the time to review our paper and provide valuable feedback. As the discussion phase is nearing its conclusion and there will be no second stage of author-reviewer interactions, we would like to confirm if our responses from a few days ago have effectively addressed your concerns. We hope they have resolved the issues you raised. If you require further clarification or have additional questions, please don’t hesitate to reach out. We are happy to continue the conversation.
> > >
> > > Thank you,
> > >
> > > The authors

---

> > > > ### Comment · Reviewer_g9ZC · 2024-11-26
> > > >
> > > > I thank the authors for their response, which addressed some of my concerns and clarified some aspects of the approach.
> > > >
> > > > However, I still strongly support that the evaluation should consider more datasets and analysis to validate the behaviour of the proposed framework, especially in the adversarial context. Thus, I will retain my score.

---

> > > > > ### Author Response · Authors · 2024-11-26
> > > > >
> > > > > We thank again the reviewer for their valuable comments and for responding to our reviews.
> > > > >
> > > > > As mentioned in the rebuttal, the dataset itself and the images in it are not used during the evaluation of the adversarial attacks. We only use the class names -- we sample two class names and use CLIP as a zero-shot classifier between them. We provide here additional results for ImageNet classes in the hope of addressing the concerns of the reviewer:
> > > > >
> > > > > We compute the success rate of the attack (as done in Table 3) for 3 random pairs of classes sampled from ImageNet (the experiment was repeated twice for each pair; we only used neurons from layer 9):
> > > > >
> > > > > | First class   | Second class   | Success Rate |
> > > > > |-------------|-------------|-------|
> > > > > | agama  | chain |   6   |
> > > > > | scorpion  | snail |   5.5   |
> > > > > | dough  | toyshop | 3.5   |
> > > > >
> > > > > As shown, the success rate is relatively similar to the success rate presented for random classes of CIFAR10.

---

### Author Response · Authors · 2024-11-21

We appreciate the time and valuable feedback provided by all the reviewers on our work. We are thankful that the paper has been positively received overall. As the discussion period is nearing its end, we kindly request that all reviewers confirm whether our rebuttal has addressed their concerns and allow us the chance to respond to any additional follow-up. Thank you once more for your participation.

---

### Meta-Review · Area_Chair_Ab2Z · 2024-12-15

**Metareview:**

The paper introduces a novel method for interpreting neurons in CLIP (Contrastive Language-Image Pretraining) through second-order effects, which measure a neuron’s influence on the model’s output via later attention layers. This approach provides a deeper understanding of individual neurons by analyzing their contributions in the joint text-image embedding space. The authors demonstrate that these second-order effects are sparse and can be approximated by single-direction vectors in the embedding space, which are then decomposed into interpretable text representations.

The proposed method is applied to two key tasks:

- Adversarial Image Generation: The polysemantic behavior of neurons is exploited to generate "semantic adversarial examples," where images fool CLIP into making incorrect classifications based on spurious correlations.
- Zero-Shot Segmentation: The second-order effects are used to identify neurons aligned with specific semantic concepts. Aggregated activation maps from these neurons are used to assign labels to image regions, enabling segmentation without additional training.

Empirical results show that the method improves segmentation performance over existing baselines (e.g., TextSpan) and demonstrates the capability of adversarial attacks that lie on the natural image manifold. The study highlights the interpretability and practical utility of second-order effects in understanding and leveraging CLIP's representations.

Overall, it is recommended that this paper be accepted.

Additional comments can be found below.

**Additional Comments On Reviewer Discussion:**

The paper itself is a good contribution to the community. However, AC agrees with the Reviewer g9ZC that extending the analysis to other datasets, e.g. CIFAR10/100, OpenImages, etc., would be great.

---

### Decision · Program_Chairs · 2025-01-22

Accept (Poster)